# Rapid biphasic decay of intact and defective HIV DNA reservoir during acute treated HIV disease

Alton Barbehenn [1,7] ✉, Lei Shi [2,7], Junzhe Shao [2], Rebecca Hoh [1], Heather M. Hartig [1], Vivian Pae [1], Sannidhi Sarvadhavabhatla [1], Sophia Donaire [1], Caroline Sheikhzadeh [1], Jeffrey Milush [3], Gregory M. Laird [4], Mignot Mathias [4], Kristen Ritter [4], Michael J. Peluso [1], Jeffrey Martin [5], Frederick Hecht [1], Christopher Pilcher [1], Stephanie E. Cohen [1,6], Susan Buchbinder [6], Diane Havlir [1], Monica Gandhi [1], Timothy J. Henrich [3], Hiroyu Hatano [1], Jingshen Wang [2], Steven G. Deeks [1] & Sulggi A. Lee [1] ✉

Despite antiretroviral therapy (ART), HIV persists in latently-infected cells (the HIV reservoir) which decay slowly over time. Here, leveraging >500 longitudinal samples from 67 people living with HIV (PLWH) treated during acute infection, we developed a mathematical model to predict reservoir decay from peripheral CD4 + T cells. Nonlinear generalized additive models demonstrated rapid biphasic decay of intact DNA (week 0-5: $t_{1/2} \sim 2.83$ weeks; week 5-24: $t_{1/2} \sim 15.4$ weeks) that extended out to 1 year. These estimates were ~5-fold faster than prior decay estimates among chronic treated PLWH. Defective DNA had a similar biphasic pattern, but data were more variable. Predicted intact and defective decay rates were faster for PLWH with earlier timing of ART initiation, higher initial CD4 + T cell count, and lower pre-ART viral load. In this study, we advanced our limited understanding of HIV reservoir decay at the time of ART initiation, informing future curative strategies targeting this critical time.

While antiretroviral therapy (ART) is able to suppress the virus to undetectable levels, the virus rapidly rebounds from latently infected cells (the HIV reservoir) within weeks of ART interruption and is, thus, not a cure[1–8]. Thus, a major goal is to eradicate and/or accelerate the decay of the reservoir in order to achieve clinical remission. However, HIV cure trials to date have largely failed to demonstrate a clinically meaningful reduction in the size of the HIV reservoir and/or lead to sustained ART-free remission[9–12]. The majority of these trials have included people living with HIV (PLWH) treated during chronic infection long after reservoir establishment (i.e., several years after

initiating ART)[10,13–17]. Recent combination trials (e.g., broadly neutralizing antibodies given with ART) have yielded more promising results, and a few participants have demonstrated extended post-intervention viral control[18–20], but the mechanisms by which these participants have enhanced viral control remain unclear.

Individuals who initiate ART earlier (< 6 months after infection) are more likely to become post-treatment controllers (PTCs), demonstrating ART-free viral control after a period of initial ART suppression[21]. PLWH treated during chronic HIV often have larger reservoirs[22–29] and exhausted/dysfunctional immune responses[30–32] (due to prolonged

[1]Department of Medicine, Division of HIV, Infectious Diseases & Global Medicine, University of California San Francisco, San Francisco, CA, USA. [2]Department of Biostatistics, University of California Berkeley, Berkeley, CA, USA. [3]Department of Medicine, Division of Experimental Medicine, University of California San Francisco, San Francisco, CA, USA. [4]AccelevirDiagnostics, Baltimore, MD, USA. [5]Department of Biostatistics & Epidemiology, University of California San Francisco, San Francisco, CA, USA. [6]San Francisco Department of Public Health, San Francisco, CA, USA. [7]These authors contributed equally: Alton Barbehenn, Lei Shi. ✉e-mail: altonbarbehenn@gmail.com; sulggi.lee@ucsf.edu

**Table 1 | UCSF Treat Acute HIV Study Population**

|  | **N = 67** |
|---|---|
| Timing of ART initiation (days from date of detected HIV infection to ART start date) | 31.0 (22.0–88.5) |
| Initial CD4 + T-cell count (cells/mm³) | 505 (350–670) |
| Pre-ART plasma HIV RNA (log₁₀ copies) | 4.85 (3.69–5.65) |
| Age | 30.0 (25.5–38.0) |
| Gender (self-reported) |  |
| Male | 65 (97.0%) |
| Cisgender Female | 1 (1.50%) |
| Transgender Female | 1 (1.50%) |
| Race/ethnicity (self-reported) |  |
| White | 22 (32.8%) |
| Latinx | 20 (29.9%) |
| Asian | 14 (20.9%) |
| Black | 10 (14.9%) |
| Other | 1 (1.5%) |
| Prior pre-exposure prophylaxis (PrEP) | 29 (43.3%) |
| HIV acquisition/PrEP overlap <10 days | 15 (22.4%) |
| PrEP was initiated but already acquired HIV | 8 (11.9%) |
| HIV acquired on PrEPᵃ | 6 (9.0%) |
| Referral HIV testing sites |  |
| San Francisco Department Public Health (%) | 28 (41.8%) |
| Community-Based Organization (%) | 29 (43.2%) |
| Private Health Clinics (%) | 10 (14.9%) |

Medians (with interquartile ranges) or frequencies (with percentages) are shown.
ᵃFor participants with HIV acquired on PrEP: median baseline plasma log₁₀HIV RNA was 2.2 copies/mL.

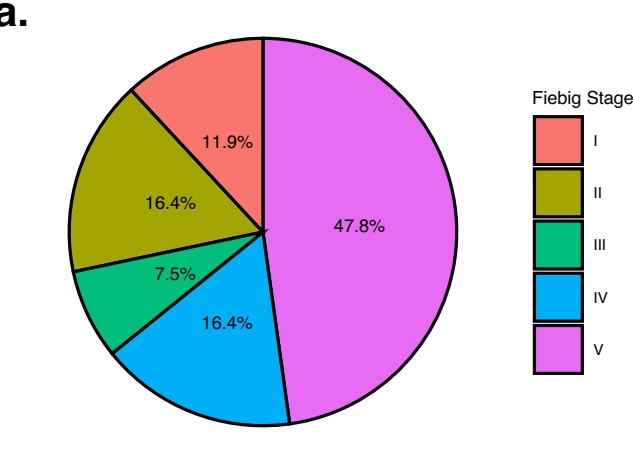

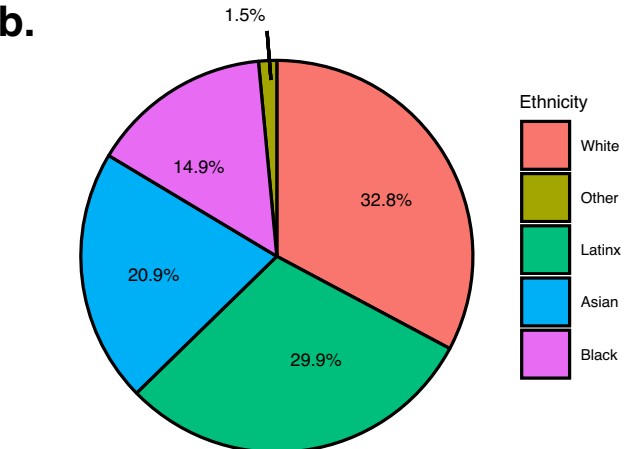

**Fig. 1 | The distribution of study participants in the UCSF Treat Acute HIV cohort.** A total of 67 participants met inclusion criteria for acute HIV, defined as <100 days since the estimated date of detected HIV infection (EDDI) using the Infection Dating Tool (https://tools.incidence-estimation.org/idt/); these estimates were then used to estimate acute HIV Fiebig stages (**a**)[44,45]. The majority of the cohort was of non-White self-reported race/ethnicity, consistent with national trends for people incident acute HIV (**b**)[46].

periods of untreated HIV infection). Thus, different host factors, such as the timing of ART initiation, initial CD4 + T cell count, or pre-ART HIV viral load, may have a profound impact on HIV reservoir decay rates, and yet there are limited reservoir decay modeling studies accounting for these factors. While there have now been a handful of studies modeling how quickly the HIV reservoir decays during prolonged ART (~20 years)[33–36], there have been fewer studies modeling decay rates after acute treated HIV[37–39], and none directly performing mathematical modeling of HIV intact and defective DNA decay.

Here, leveraging >500 longitudinal blood samples, we developed a mathematical model of reservoir decay among 67 participants from the UCSF Treat Acute HIV cohort initiating ART <100 days of HIV infection. We fit various mono-, bi-, and triphasic decay curves for both HIV intact (infected cells harboring intact viral sequences able to produce infectious virions) and defective (the majority of the HIV reservoir but incapable of producing infectious virions) DNA, and we observed biphasic decay patterns for both measures. Furthermore, both HIV intact and defective DNA decay rates were significantly faster among PLWH with known clinical factors associated with enhanced host viral control: higher initial CD4 + T cell count, earlier initiation of ART, and lower pre-ART viral loads[22,23,33,40,41]. As further validation of our mathematical modeling approach, we also fit decay models for plasma HIV RNA (viral load measured at each study visit using a standard clinical assay with a limit of detection < 40 copies/mL). We observed a triphasic decay of plasma HIV RNA, similar to prior reports among PLWH-initiating ART[3,4,39].

## Results
### Characteristics of study participants
A total of 67 adults (83% of those screened) with a new diagnosis of acute HIV (<100 days between HIV infection to ART initiation date) were included in the study (Table 1 and Supplementary Fig. 1). All 67 participants completed monthly follow-up visits in the study for the full 24 weeks. A large proportion (65.7%) of participants were co-

enrolled in our longitudinal UCSF SCOPE HIV cohort and remained in the study beyond 24 weeks, with study visits approximately every 3-4 months. The median follow-up for our cohort was 0.81 (interquartile range = 0.47–1.66) years. We calculated the estimated date of detected infection (EDDI) for each participant using an algorithm[42,43] successfully applied to other acute HIV cohorts[37,38] (Supplementary Fig. 2). We also estimated Fiebig stage[44,45] for each participant, an older but often cited method for staging recency of HIV infection (Fig. 1). Consistent with our San Francisco-based study population, participants were mostly male (97%) and reflected local and national racial/ethnic trends of higher incident acute HIV in these populations (Fig. 1)[46]. Baseline study visits HIV-1 antigen/antibody (Architect) and HIV-1 antibody (Geenius) testing demonstrated 27% and 28% false negative/indeterminate rates (Supplementary Fig. 3), respectively, consistent with our San Francisco Department of Public Health (SFDPH) reported estimates for new acute HIV diagnoses[47]. Genotype data (Monogram) were available for a subset of 57 participants; 77% had wild-type HIV, 9% had M184V/I mutations (all were reported among participants citing prior and/or current pre-exposure prophylaxis [PrEP] use), and 14% had evidence of possible partner-transmitted resistance mutations (based on referral of newly diagnosed partners within our cohort and/ or SFDPH partner tracing[47]).

Our cohort also reflected a high proportion of self-reported prior PrEP use (42% ever use, 20% use in the past 10 days), reflecting San Francisco's early and widespread adoption of PrEP[47]. All PrEP reported in this study was oral PrEP with tenofovir disoproxil fumarate/emtricitabine (TDF/FTC), as this was the only form clinically available during the study period. Among individuals reporting overlapping PrEP use within 10 days of their EDDI, six participants had probable HIV acquisition while on PrEP (median baseline $\log_{10}$HIV RNA = 2.2 copies/mL, ~3 $\log_{10}$ lower than those not reporting PrEP overlap) (Table 1 and Supplementary Fig. 4), including one participant[48] who may have

acquired HIV in the setting of therapeutic PrEP concentrations (confirmed by plasma and hair ART concentrations).

## Rapid biphasic decay of HIV intact and defective DNA

Overall, after fitting various mono-, bi-, and triphasic decay curves using semiparametric generalized additive models, we found that a biphasic decay pattern with an inflection point ($\tau$) = week 5 best fit the data for HIV intact and defective DNA (Table 2, Figs. 2, 3 and Supplementary Fig. 5). Validation of these models against the observed data showed good model performance (Fig. 4 and Supplementary Figs. 6, 7) and that HIV intact and defective DNA decay patterns significantly predicted faster decay rates (Figs. 5, 6) for participants with known clinical factors associated with smaller HIV reservoir size[22,23,33,40,41].

First, we modeled HIV intact and defective DNA using a linear effect of time on ART (which assumes a constant rate of change regardless of the duration of viral suppression). However, since we observed evidence of nonlinearity, we fit nonlinear generalized additive models to better estimate HIV intact and defective DNA decay patterns. For all models, we tested clinical factors of age, pre-ART CD4 + T cell count, pre-ART viral load, and timing of ART initiation for inclusion as potential covariates. We found that both HIV intact and effective DNA were well described by a biphasic model, comparing

**Table 2 | Prediction performance of monophasic, biphasic, and triphasic generalized additive models of HIV reservoir decay during weeks 0–24**

| HIV Reservoir | Monophasic (95% CI) | Biphasic (95% CI) | Triphasic (95% CI) |
|---|---|---|---|
| Intact DNA | 886 (722, 1023) | 797 (595, 968) | 796 (597, 965) |
| Defective DNA | 1426 (1344, 1504) | 1268 (1183, 1343) | 1272 (1188, 1348) |

We performed bootstrapping to estimate the Akaike information criteria (AIC) value and 95% confidence intervals for monophasic, biphasic, and triphasic models for both HIV intact and defective DNA assays.

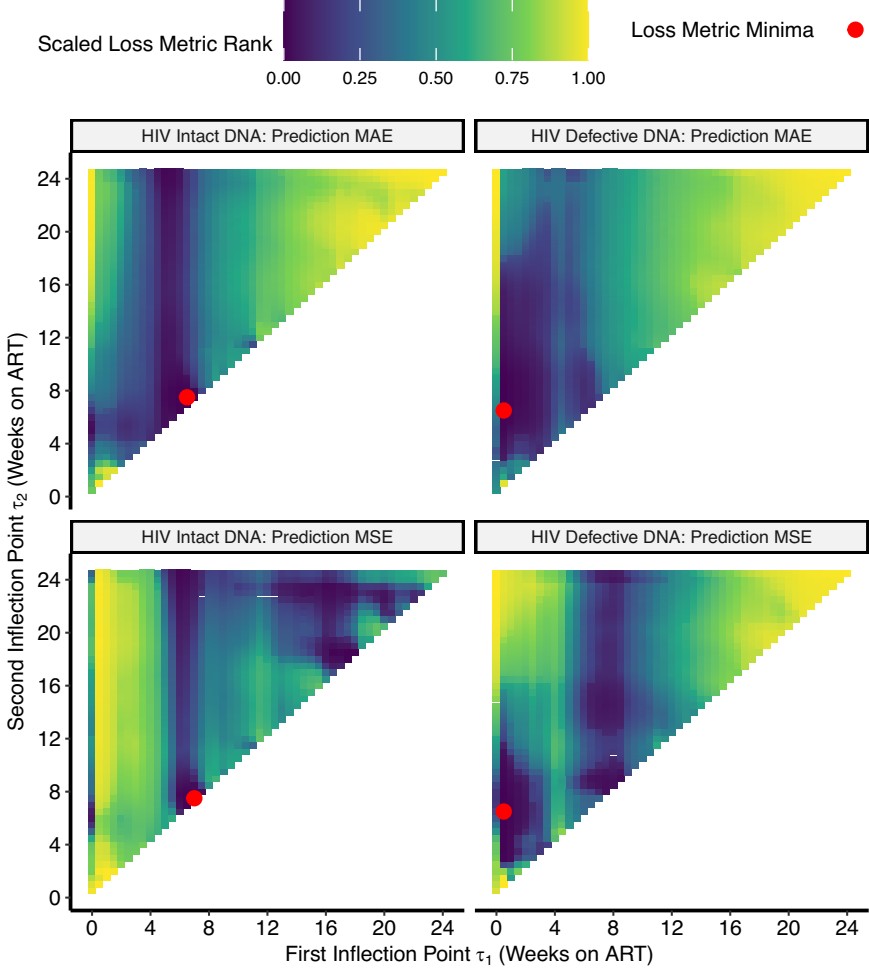

**Fig. 2 | Determination of optimal inflection points for HIV intact and defective DNA triphasic decay models.** Using the triphasic models for HIV intact (left panel) and defective (right panel) DNA, we then determined the optimal inflection points, $\tau$, by minimizing the predicted mean absolute error (MAE; top panels) using leave-one-out cross-validation or the predicted mean squared error (MSE; bottom panels). Red dots denote the optimal inflection points, $\tau$, for each model and

prediction loss metric. For HIV intact DNA, the first (x-axis) and second (y-axis) inflection points were relatively similar, suggesting that a single inflection point – i.e., a biphasic model – adequately described the data. For HIV defective DNA, the first inflection point (x-axis) was close to zero, this again suggested that a biphasic model reasonably described the data.

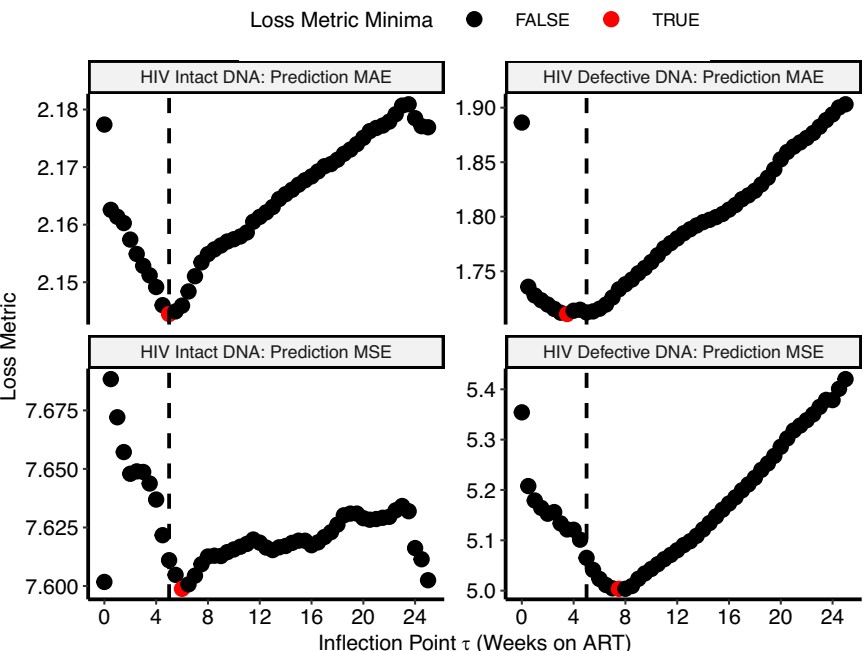

**Fig. 3 | Determination of optimal inflection points for HIV intact and defective DNA biphasic decay models.** Using the biphasic models for HIV intact (left panel) and defective (right panel) DNA, we then determined the optimal inflection point, $\tau$, by minimizing the predicted mean absolute error (MAE; top panels) using leave-one-out cross-validation or the predicted mean squared error (MSE; bottom panels). An inflection point of $\tau = 5$ weeks (vertical dashed line) best-fit decay patterns for both HIV intact (left panels) and defective (right panels) DNA. Red dots denote the best $\tau$ for each model and prediction error metric.

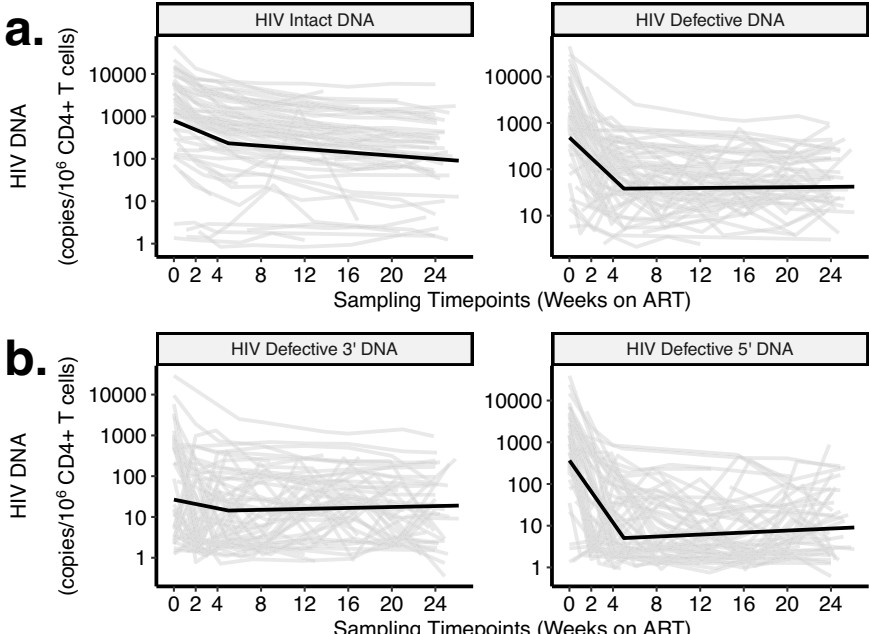

**Fig. 4 | Predicted decay patterns of HIV intact and defective DNA during acute treated HIV from weeks 0–24.** Decay patterns for observed (thin gray lines) HIV intact and total defective (**a**), as well as 3′ and 5′ defective (**b**) DNA closely fit with average model predictions (thick black lines). Sampling time points are labeled on the x-axis (including a week 2 study visit during which confirmatory HIV test results were disclosed). We estimated average predicted participant decay rates by taking the mean of $E_i$ (estimated time between HIV infection and ART initiation), $C_i$ (initial CD4 + T cell count), and $V_i$ ($\log_{10}$ pre-ART plasma viral load) across participants from final models.

Akaike information criteria (AIC) (Table 2) and thus was chosen over a triphasic model since comparing the minimum predicted mean absolute error (MAE) using leave-one-out cross-validation and/or the minimum predicted mean squared error (MSE) (Fig. 2), suggested similar inflection points. For HIV intact DNA, the first and second inflection points were similar, suggesting that a single inflection point – i.e., a biphasic model – adequately described the data, and for HIV defective DNA, since the first inflection point was close to zero, this again suggested that a biphasic model well described the data. We then further determined that the inflection point of $\tau = 5$ weeks, after

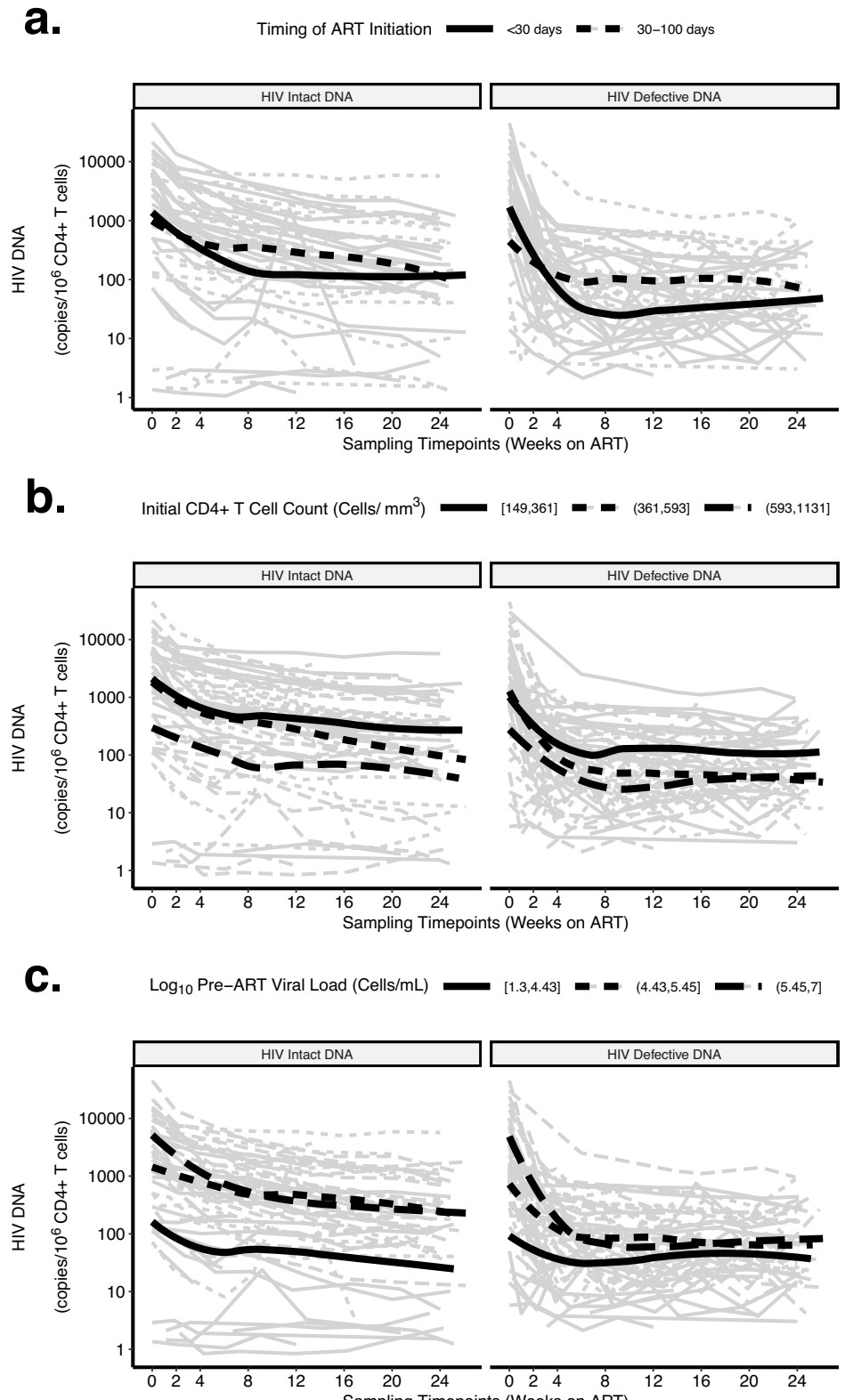

**Fig. 5 | HIV intact and defective DNA decay patterns were associated with known clinical factors associated with HIV reservoir size.** The observed HIV DNA data are shown as thin gray lines for each participant, while the decay pattern for the model-predicted average participant is shown as thick black lines. Biphasic decay patterns for HIV intact (left panel) and combined defective (3' plus 5', right panel) were faster among participants initiating ART earlier (< 30 days vs. 30–100 days) (**a**), with higher initial CD4 + T cell counts (shown by tertiles) (**b**), and lower pre-ART viral load (shown by tertiles) (**c**).

comparing MAEs and MSEs, was optimal for both HIV intact and defective DNA (Fig. 3 and Supplementary Fig. 5). Since we found that several key clinical factors (previously associated with HIV reservoir size initiation[22,23,33,40,41]) were strongly associated with HIV DNA decay rates (Figs. 5, 6), all final models included terms for initial CD4 + T cell count, pre-ART viral load, and timing of ART initiation.

Our final biphasic decay model of HIV intact DNA demonstrated a rapid $t_{1/2} \sim 2.83$ (95%CI = 2.39–3.27) weeks for the first ~ 5 weeks of AR,

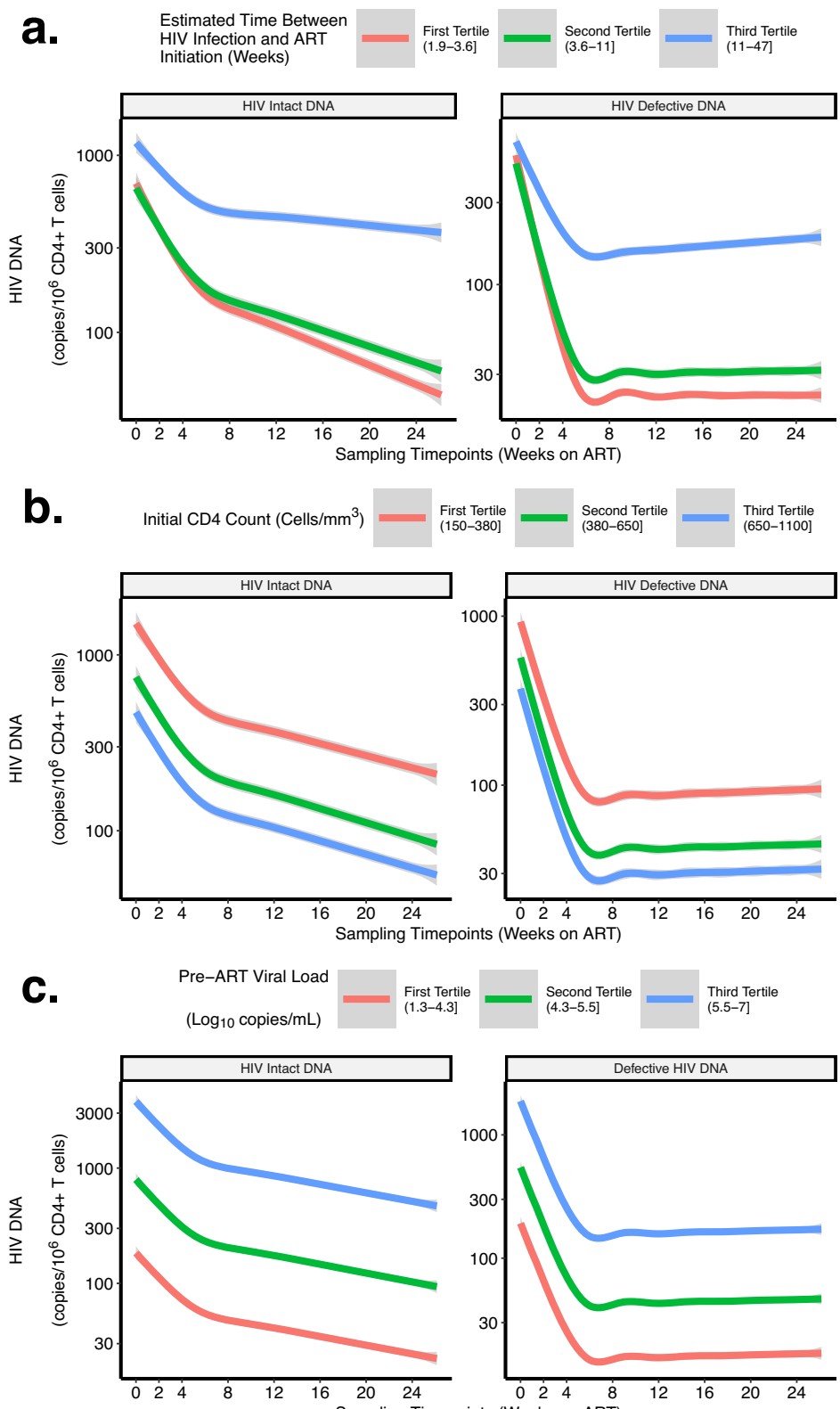

**Fig. 6 | Predicted HIV intact and defective DNA decay rates, by tertiles of clinical factors associated with HIV reservoir size.** We performed bootstrapping to estimate the average predicted decay rates of HIV intact (left panels) and defective (right panels) DNA, stratified by tertiles of known clinical factors associated with HIV reservoir size: timing of ART initiation (**a**), initial CD4 + T cell count (**b**), and pre-ART viral load (**c**). Figures depict the bootstrapped mean and its 95% confidence interval.

followed by a slower second decay phase with a $t_{1/2} \sim 15.4$ (95% CI = 12.0–21.9) weeks (Supplementary Table 1). HIV defective DNA had a similar pattern, with an initial rapid decay ($t_{1/2} \sim 1.36$, 95% CI = 1.17–1.55 weeks), followed by a slower decay, but the change in decay was not statistically significant given the large variability in HIV defective DNA during this second phase (Fig. 4). Interestingly, we observed a significantly faster decay of HIV defective vs. intact DNA during the first phase ($p < 1e\text{-}16$) (Fig. 4). While the reasons for this are unclear, given the frequency of our sampling at these acute HIV timepoints, our observation may potentially be due to (1) a true biological phenomenon uniquely captured by our frequent early sampling and/or (2) reflect unique properties of the IPDA (see Discussion).

Our final models also demonstrated significantly faster decay rates with clinical factors associated with smaller reservoir sizes (Figs. 5, 6). For example, our models estimated that for HIV intact DNA, for each week earlier that ART was initiated, the $t_{1/2}$ was predicted to be reduced by $\sim 0.0827$ (95%CI = 0.0203-0.145) and by $\sim 1.08$ (95% CI = 0.316–1.84) during the second phase (Supplementary Table 3). Similarly, our models predicted that higher initial CD4 + T count and lower pre-ART HIV RNA predicted significantly faster HIV intact and defective decay rates (Supplementary Table 4). Further validation using fitted spline models again demonstrated that higher initial CD4 + T count and lower pre-ART HIV RNA predicted faster HIV intact and defective DNA decay rates (Supplementary Fig. 10). For example, a participant with an initial CD4 + T cell count of 900 cells/mm³ was predicted to have $\sim 10$ times faster decay of HIV intact DNA than a participant with an initial CD4 + T cell count of 300 cells/mm³. Similar patterns were observed for HIV defective DNA, but the fitted splines were less linear.

While were unable to perform adjusted analyses for other important clinical factors such as gender and race/ethnicity, given the small sample sizes in our study (Table 1), we did perform sensitivity analyses focusing on the small number of cisgender and transgender women, as well as the small numbers of PLWH reporting PrEP use within 10 days of HIV diagnosis. These analyses demonstrated that results were overall relatively unchanged and that these participants did not necessarily fall in the lower range of reservoir measurements (Supplementary Fig. 8). Furthermore, to ensure that the selected inflection point of $\tau = 5$ weeks was not influenced by potential outlier data, we performed three different sensitivity analyses excluding participants for whom HIV reservoir measures might fall on the higher and/or lower range of values: (1) individuals reporting prior PrEP use within 10 days of HIV infection, (2) participants with plasma viral load blips (defined as a one-time viral load >1000 copies/mL or two consecutive viral loads >100 copies/mL between weeks 0–24), and (3) participants with sudden increases in HIV intact DNA (defined as > 50% increase between two consecutive measurements during weeks 0–24). These sensitivity analyses demonstrated that $\tau = 5$ weeks remained a reasonable choice for the model's inflection point (Supplementary Fig. 9) and that the estimates were overall unchanged after exclusion (Supplementary Table 2).

### Triphasic decay of plasma HIV RNA
As further validation of our mathematical modeling approach, we also fit decay models for plasma HIV RNA. Plasma HIV RNA (viral load) was measured at each study visit using a standard clinical assay (Abbott Real-Time PCR assay, limit of detection < 40 copies/mL). We again fit various mono-, bi-, and triphasic models, and a triphasic decay model best fit these data with inflection points at 0.5 and 4 weeks (Supplementary Fig. 11). Our findings are consistent with prior published work describing triphasic decay of plasma HIV RNA in treatment naïve PLWH initiating integrase inhibitor-based therapy[49]. Our final adjusted models predicted a rapid initial decay ($t_{1/2} \sim 0.659$, 95% CI = 0.541-0.778 days), a second decay ($t_{1/2} \sim 4.93$, 95% CI = 3.98–5.89 days), with no significant decay during the third phase

(Supplementary Table 5), closely mirroring prior reported estimates describing a $t_{1/2} = 1.14$, 9.19, and 340 days, respectively[49]. As expected, we observed that the majority of our cohort had undetectable plasma viremia by a median of 4.14 weeks, consistent with viral suppression rates among treatment naïve PLWH initiating integrase inhibitor-based ART[50,51]. Similar to the approach used for our HIV DNA decay models, we validated our final triphasic plasma HIV RNA decay model by comparing predicted vs. observed values and found that the model produced unbiased estimates across a range of plasma HIV RNA values (Supplementary Fig. 12). We again found that known clinical factors associated with reservoir size –e.g., initial CD4 + T cell count and earlier timing of ART initiation –were also associated with accelerated decay rates (Supplementary Figs. 13–15).

## Discussion
Leveraging > 500 longitudinal blood samples from the UCSF Treat Acute HIV cohort, we performed mathematical modeling and demonstrated a rapid biphasic decay of HIV intact and defective DNA. Our estimates for HIV intact DNA decay were significantly faster ($\sim 5$-fold) compared to prior estimates from chronic treated[3] PLWH initiating ART. Furthermore, clinical factors associated with smaller HIV reservoir sizes (e.g., earlier timing of ART initiation, higher initial CD4 + T cell count, and lower pre-ART viral load) predicted faster decay rates of both HIV intact and defective DNA. We further validated our modeling approach by fitting plasma HIV RNA decay rates, and we observed a triphasic decay pattern, consistent with prior estimates[49]. Our mathematical modeling approach may serve as a meaningful way to predict expected decay rates after ART initiation and the potential impact of clinical factors that may differ when comparing across global HIV cohorts that may also have different host genetics, HIV-1 subtypes, etc. This approach may also help inform the design of future HIV cure trials, e.g., to predict optimal timeframes during which an intervention may have the greatest impact on accelerating reservoir decay and/or limiting reservoir establishment.

Our findings compare to several key prior modeling studies of HIV reservoir decay[3,38,52–54], all of which fit mostly unadjusted fully parameterized mixed effects models but still lend support to our findings. For example, we observed an initial rapid HIV intact DNA decay rate of $t_{1/2}$ of $\sim 2.83$ weeks ($\sim 0.71$ months), followed by a slower second phase with a $t_{1/2} \sim 15.4$ weeks ($\sim 3.9$ months). Strikingly, this first phase decay estimate is nearly identical to prior reports in chronic treated PLWH initiating ART ($t_{1/2} = 0.43$ months)[3], but our estimates for the second phase of decay were $\sim 5$-fold faster than estimates from this other study ($t_{1/2} = 19$ months), well below their confidence limits (8.23–43.7 months).[3] Our faster rate of HIV intact DNA decay during this second phase is unclear but may potentially be due to true biological differences (e.g., less exhausted immune cells compared to chronically treated PLWH[55,56]) or reflect greater precision in estimating decay rates from our frequent sampling (every 2–4 weeks). The initial rapid decay of HIV-infected cells after ART initiation is thought to be largely due to clearance of free virions and death of productively infected cells[3,4,49,57,58]. We estimated similar first-phase decay rates as those previously reported in chronically treated PLWH initiating ART[3], suggesting that death of productively infected cells, regardless of the timing of ART initiation, may indeed be driving the first-phase decay estimates. Furthermore, plasma (cell-free) HIV RNA correlates with the frequency of productively infected CD4 + T cells[59]; our plasma HIV RNA decay estimates provide further support as these estimates are again consistent with prior reported clearance rates of productively infected CD4 + T cells ($t_{1/2} \sim 0.7$ days)[60]. Meanwhile, the second phase of reservoir decay after ART initiation is thought to represent a contraction phase when activated cells transition from an effector to a memory phenotype with ART-mediated antigen reduction[61–63]. This second phase is thought to be largely driven by the death of longer-lived memory cells[64,65]. Indeed, if we extrapolate the second phase of

our HIV intact DNA model, we estimate that PLWH who delay ART initiation to ~56 weeks after HIV infection have a predicted $t_{1/2}$ that is comparable to the (slower) second phase decay reported in chronically treated PLWH[3]. Our data suggests that – especially during this second phase of decay – curative interventions given during this critical window of time may have the potential to significantly reduce the establishment of these long-lived memory cells.

Our data are also consistent with findings from two prior acute HIV cohorts[37,38] that did not measure IPDA (HIV intact and defective DNA) but did measure HIV total, integrated DNA, and 2-LTR DNA by real-time PCR[66] and also performed the quantitative viral outgrowth (QVOA[24,64,67]) and multiply spliced tat/rev (TILDA[68]) assays. One of these studies by Massanella and colleagues performed mathematical modeling and also demonstrated biphasic decays of HIV total, integrated, and 2-LTR DNA, with a similar inflection point (6 weeks)[38]. While they were unable to report decay models for QVOA or TILDA (likely due to the low frequency of HIV-infected cells despite the large number of input cells[69,70], which may have precluded more complex decay modeling), their estimates for HIV total, integrated, and 2-LTR decay rates closely compare to our estimates for defective DNA. The population of HIV-infected cells generally falls into three broad categories: (1) truly intact proviruses, (2) partially defective proviruses that can produce defective HIV RNA/proteins, which, despite being unable to produce virus, can still lead to immunogenic/cytopathic effects[71], and (3) truly inert proviruses that express no HIV RNA or proteins. While the assays in this other study did not specifically discriminate intact from defective viral sequences, since the majority of the HIV reservoir consists of defective provirus[69] and since the majority of infected cells in acute PLWH consist of these highly unstable unintegrated linear HIV DNA (with an estimated half-life of ~2 days)[72], the overlap in our modeling results may suggest an overlap in the population of HIV-infected cells captured by our respective assays.

Finally, it is important to note that the decay rates described here are likely complementary to, but not the same as, decay rates described in several long-term ART studies[33–36,64,67]. First, these long-term ART studies (in chronically treated PLWH) did not sample participants at the time of ART initiation and had less frequent sampling over longer periods of ART suppression[33–35,54]. Overall, these studies described a biphasic decay (inflection point ~7 years of ART) with a $t_{1/2}$ ~44 months for HIV intact DNA[64,67] and found that HIV intact DNA decayed faster than defective DNA, presumably due to preferential clearance of intact, or replication-competent, provirus during long-term ART[33–35,54,73,74]. However, HIV intact DNA decay rates have also been shown to plateau or even increase in some individuals during prolonged ART[35,54]. Our biphasic model identified a somewhat surprising finding that HIV- defective DNA decayed faster than HIV-intact DNA during the first phase. The reasons for this are unclear but may reflect true biological phenomena uniquely captured by our frequent early sampling and/or unique properties of the IPDA. Since the majority of the HIV reservoir consists of defective provirus[69], estimates of HIV total, integrated, and 2-LTR DNA decay rates from the study by Massanella et. al., ($t_{1/2}$ = 14.5, 14.1, and 30.5 days, respectively[38]) are largely consistent with our estimates of defective DNA decay rates ($t_{1/2}$ = 9.5 days) during the first phase of decay, suggesting potential true biological phenomena that warrant further study. Alternatively, a second possibility is that our observations reflect some misclassification of HIV intact provirus (i.e., since the IPDA targets just two regions of the HIV genome to define defective provirus[75]). However, Reeves and colleagues recently performed detailed validation experiments (e.g., using quantitative viral outgrowth assay and near full-length sequencing) and showed that the rate of misclassification is < 5% with the IPDA[52], suggesting that this degree of misclassification alone would be unlikely to fully explain our findings.

Our study has several limitations that deserve mention. While we leveraged several hundred longitudinal blood samples from acutely treated PLWH, we did not model the HIV tissue reservoir; our tissue studies are currently underway but will be limited in the number of longitudinal time points to perform similar detailed modeling. Since the peripheral HIV reservoir largely reflects proviruses originating from the tissue reservoir[37,76–78], tissue reservoir decay estimates in ours, as well as other studies, should be modeled in parallel with the more frequently sampled peripheral reservoir decay estimates in future work. We performed IPDA, which, while highly scalable for a large number of samples, less accurately quantifies the replication-competent reservoir compared to near-full-length proviral sequencing or QVOA. Nonetheless, HIV intact DNA measured by IPDA closely reflects results from these other assays, even considering the known enrichment of integrated forms of HIV DNA observed in acute PLWH[37]. As with all molecular assays for HIV, certain polymorphisms at primer or probe binding sites can impact IPDA assay performance. We observed IPDA signal failure for 6 participants (8.9%) – a rate consistent with reports from large HIV cohorts from North America and Europe where subtype B predominates (6-7%)[33,79]. We also did not measure changes in the clonal landscape (e.g., HIV integration). The clonal landscape at the time of acute HIV is very diverse, and we hypothesize that this effect is more likely to have a greater impact after a longer duration of ART suppression. Future models should include these parameters to formally test this hypothesis. Finally, there are few highly characterized acute HIV cohorts to date, and each study possesses unique host and viral characteristics making direct cross-cohort comparisons challenging. Our study included mostly men who have sex with men and HIV-1 subtype B. It will be critical to validate our HIV reservoir decay models in global populations with distinct host genetic ancestry, HIV-1 subtypes, and clinical features to facilitate cross-cohort comparisons and inform future HIV cure trial design and interpretation.

The long-lived latent reservoir is a key defining target for HIV cure, but how and where these cells then become the long-lasting latent reservoir remains unclear. Even in reservoir decay studies analyzing data out to 20 years of ART suppression, decay patterns are not broadly generalizable[35,54]. Thus, there is a critical need for a scalable approach to broaden our understanding of HIV reservoir decay patterns across a global population of PLWH, ideally aligning study designs and assays and performing meta-analyses, including how key clinical factors such as the timing of ART initiation, initial CD4 + T cell count, and pre-ART viral load influence decay rates.

## Methods

### Study participants

Individuals with newly diagnosed acute (<100 days) HIV infection were enrolled in the UCSF Treat Acute HIV cohort between December 1, 2015, and November 30, 2020, and co-enrolled in the UCSF SCOPE HIV cohort, an ongoing longitudinal study of over 2500 PLWH. Eligible participants were provided same-day ART initiation with tenofovir/emtricitabine (TDF/FTC, then TAF/FTC once available in 2016) + dolutegravir (DTG) and linked to clinical care[47]. Individuals reporting concomitant PrEP use (<100 days from any potential exposure to HIV by history and/or clinical test results) were also started on darunavir+ritonavir (DRV/r) as a fourth drug, which was continued until confirmation of baseline HIV genotyping test results (Monogram Biosciences, South San Francisco, CA, U.S.A.). Additional ART changes necessary for clinical care (e.g., laboratory abnormalities, drug-drug interactions, and/or participant preference) were honored and adjusted during the period of study. Participants signed a release of information that allowed clinical data extraction to determine prior HIV negative test results from the SFDPH, as well as additional HIV test results.

Study participants were seen for monthly study visits for the first 24 weeks (including an additional week 2 visit to confirm HIV test results from baseline visit) and then every 3-4 months thereafter.

Inclusion criteria for the study were prior HIV-negative testing within the last 90 days, laboratory-confirmed HIV-1 infection by antibody/antigen and/or plasma HIV RNA assay, and willingness to participate in the study for at least 24 weeks. Participants with severe renal or hepatic impairment, concurrent treatment with immunomodulatory drugs, or exposure to any immunomodulatory drugs in the preceding 90 days prior to study entry, pregnant or breastfeeding women, or participants unwilling to agree to the use a double-barrier method of contraception throughout the study period, were excluded. For each study participant, the estimated date of detected HIV infection (EDDI) was calculated using the Infection Dating Tool (https://tools.incidence-estimation.org/idt/)[42]. At each visit, detailed interviews included questions regarding current medications, medication adherence, intercurrent illnesses, and hospitalizations were performed. In addition, peripheral blood sampling at each visit was performed to measure plasma HIV RNA (Abbott Real-Time PCR assay, limit of detection < 40 copies/mL), CD4 + T cell count, and clinical labs (complete blood count, metabolic panel). All participants provided written informed consent, and the institutional review board of UCSF approved the research.

## HIV reservoir quantification

The frequencies of HIV intact and defective (3' and 5') DNA were quantified using the intact proviral DNA assay (IPDA)[80]. CD4 + T cells were isolated from cryopreserved PBMCs (EasySep Human CD4 + T cell Enrichment Kit, Stemcell Technologies), with cell count, viability, and purity assessed by flow cytometry. Negatively selected CD4 + T cells were recovered (median cells = 2x10$^6$ with median viability = 97%), and genomic DNA was extracted using the QIAamp DNA Mini Kit (Qiagen). DNA concentration and quality were determined by fluorometry (Qubit dsDNA BR Assay Kit, Thermo Fisher Scientific) and ultraviolet-visible (UV/VIS) spectrophotometry (QIAxpert, Qiagen). The frequency of intact provirus was determined using two multiplex digital droplet polymerase chain reaction (ddPCR) assays performed in parallel: (1) the HIV-1 Proviral Discrimination reaction, which distinguishes intact from defective provirus via two strategically placed amplicons in HIV psi and RRE regions as well as a hypermutation discrimination probe, and (2) the Copy Reference/Shearing reaction, which quantifies DNA shearing and input diploid cell equivalents using the human *RPP30* gene[80]. All ddPCR reactions were assembled via automated liquid handles to maximize reproducibility and analyzed using the BioRad QX200 AutoDG Digital Droplet PCR system (BioRad). Up to 700 ng of genomic DNA were analyzed per reaction, and final input DNA concentrations were dependent upon recovered DNA concentrations. Samples were batch processed and analyzed, including negative controls from uninfected donors and J-Lat full-length clone 6.3 (E. Verdin, Gladstone Institutes and UCSF, San Francisco, CA, USA) cells as positive controls. Across > 500 IPDA measurements, we interrogated a median of 4.8 x 10$^5$ CD4 + T cell genomes per assay and observed a median DNA shearing index (DSI) of 0.40.

## Statistical methods

We developed a semiparametric biphasic decay model to estimate the HIV DNA reservoir size over time in log$_{10}$ copies per 10$^6$ CD4 + T cells as

$$\log_{10}(I_{it}) \sim f_1(T_{it}; \tau, \beta_1, \beta_2) + E_i \cdot f_1(T_{it}; \tau, \beta_3, \beta_4) + f_2(C_i) + f_3(V_i) + \mu_i \tag{1}$$

where $I_{it}$ represents either the HIV intact or defective DNA reservoir size of the $i$-th participant at $t$-th visit. The number of weeks since ART initiation is denoted $T_{it}$. The model additionally accounts for baseline clinical information defined as the initial CD4 + T cell count, $C_i$, pre-ART viral load, $V_i$, and the estimated time between HIV infection and ART initiation, $E_i$. The delay in ART initiation was centered to have a mean of zero prior to analysis; this offset was

~ 60 days in our cohort. Participant-level random effects, $\mu_i$, are also included. Building on existing models[3,54], we parameterized the decay as a continuous, linear spline with a single knot at $\tau$: $f_1(T_{it}; \tau, \beta_1, \beta_2) = \beta_1 \cdot \min\{T_{it}, \tau\} + \beta_2 \cdot \max\{T_{it}, \tau, 0\}$. Under this parameterization, $\beta_1$ and $\beta_2$ represent the decay rate before and after $\tau$, respectively. For triphasic models, the decay was modeled as a continuous, linear spline with knots at $\tau_1$ and $\tau_2$. For monophasic models, the decay was modeled as a linear function of time. The same spline parameterization and inflection point(s) were used to model the time on ART and the interaction between time on ART and delay in ART initiation; different slopes were estimated for these two terms. Cubic splines were used for both $f_2(C_i)$ and $f_3(V_i)$. After fixing the inflection point(s), model estimation was performed using the mgcv (v1.9-1) package in R (4.3.1). A two-sided Welch's $t$ test was used to compare decay rate estimates across models.

Regardless of the HIV measure (intact DNA, defective DNA, or plasma RNA), the inflection point, $\tau$, was estimated by minimizing the model's mean absolute prediction error. Candidate $\tau$ values were tested iteratively (from 0 weeks to 24 weeks), and the mean absolute errors (MAEs) were estimated using leave-one-out cross-validation:

$$MAE(\tau) = \sum_{i,t} \left| \log_{10}(I_{it}) - \log_{10}\left(\hat{I}_{it}^{(-i)}\right) \right| \tag{2}$$

where $\hat{I}_{it}^{(-i)}$ reflected the predicted HIV DNA counts for participant $i$ at time $t$ using the model fit for each participant (excluding participant $i$). Inflection points for the triphasic model were estimated similarly. We then compared the fit of various models using Akaike information criteria (AIC). Model performance metrics from cross-validation and bootstrapping are in Supplementary Data 1.

To facilitate the interpretability of our results and to allow direct comparison with prior reports[3,38,49], we estimated decay half-lives and their confidence intervals for each phase of decay, using the multivariate delta method[81]. For example, the half-life in the first decay phase, from model (1), was calculated as

$$t_{1/2}(E_i) = -\frac{0.25 \log_{10}(2)}{\beta_1 + \beta_2 E_i} = -\frac{0.25 \log_{10}(2)}{\beta_1} + \frac{0.25 \log_{10}(2) \beta_2 E_i}{\beta_1^2} + O(E_i^2) \tag{3}$$

where the second equality reflected the degree-one Taylor series about $E_i = 0$. We centered $E_i$ prior to model estimation to justify the degree-one Taylor series approximation of half-life. Thus, we estimated the baseline t$_{1/2}$ as $g_1(\beta_1, \beta_3) = -0.25 \log_{10}(2)/\beta_1$ and the adjusted t$_{1/2}$ (for each week delay in ART initiation) as $g_2(\beta_1, \beta_2) = 0.25 \log_{10}(2) \beta_2/\beta_1^2$. Finally, we included our estimated model parameters in the delta method to obtain half-life estimates:

$$\sqrt{n}\left( \begin{bmatrix} g_1(\hat{\beta}_1, \hat{\beta}_2) \\ g_2(\hat{\beta}_1, \hat{\beta}_2) \end{bmatrix} - \begin{bmatrix} g_1(\beta_1, \beta_2) \\ g_2(\beta_1, \beta_2) \end{bmatrix} \right) \rightarrow^d N\left( \begin{pmatrix} 0 \\ 0 \end{pmatrix}, \mathbf{J\Sigma J}^T \right) \tag{4}$$

$$\mathbf{J} = 0.25 \log_{10}(2) \begin{bmatrix} \frac{1}{\beta_1^2} & 0 \\ -2\frac{\beta_2}{\beta_1^3} & \frac{1}{\beta_1^2} \end{bmatrix} \tag{5}$$

where $\mathbf{\Sigma}$ reflected the covariance between $\beta_1$ and $\beta_2$, and $\mathbf{J}$ was the Jacobian matrix of $\mathbf{g}(\beta_1, \beta_2)$. For further interpretability, we calculated the percent decay/week (prior to $\tau$) using the transformation $h(\beta_1) = -100(2^{\beta_1} - 1)$. Similar calculations were performed for the second decay phase using $\beta_3$ and $\beta_4$ instead of $\beta_1$ and $\beta_2$.

We performed further validation of our proposed HIV DNA (intact, defective) and HIV RNA (plasma) decay models against known

clinical factors associated with HIV reservoir size[22,23,33,40,41]. Focusing on the clinical covariates of (i) initial CD4 + T cell count, (ii) pre-ART viral load, and (iii) timing of ART initiation (days from HIV infection to ART start date), we performed bootstrapping predictions by resampling and generating 300 new participants. The final HIV decay models (intact DNA, defective DNA, plasma RNA) were used to predict decay patterns for each resampled (bootstrapped) participant. For data visualization, we partitioned the resampled data into tertiles to demonstrate average predicted decay patterns by tertiles of each clinical predictor.

### Reporting summary

Further information on research design is available in the Nature Portfolio Reporting Summary linked to this article.

## Data availability

The raw clinical data are protected and are not available due to data privacy laws. De-identified processed virologic and clinical data generated in this study are available and have been deposited in the Dryad database (https://doi.org/10.5061/dryad.q573n5tsd). Source data are provided in the Supplementary Data file.

## Code availability

Code to reproduce our analyses and figures is provided in an R markdown file available through the Zenodo database (https://doi.org/10.5281/zenodo.13887677).

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

## Acknowledgements

The authors wish to acknowledge the participation of all the study participants who contributed to this work as well as the clinical research staff of the UCSF Treat Acute HIV and SCOPE cohorts who made this research possible. This work was supported in part by the National Institutes of Health: K23GM112526 (S.A.L.), the DARE Collaboratory (UM1AI164560; S.G.D.)., the AccelevirDx HIV Reservoir Testing Resource (U24AI143502; G.M.L.), and NIH/NIAID R01A141003 (T.J.H.). This work was also supported by the amfAR Research Consortium on HIV Eradication, a.k.a. ARCHE (108072-50-RGRL; S.G.D.), the Bill & Melinda Gates Foundation (INV-002703; S.G.D.), and investigator-initiated research grants from ViiV Healthcare (A126326; S.A.L.) and Gilead Sciences (IN-US-236-1354; S.A.L.). The content of this publication does not necessarily reflect the views or policies of the Department of Health and Human Services, the San Francisco Department of Health, nor does mention of trade names, commercial products, or organizations imply endorsement by the U.S. Government. The funders had no role in study design, data collection and analysis, decision to publish, or preparation of the manuscript.

## Author contributions

All authors provided critical feedback in finalizing the report. S.G.D., H.H., and S.A.L. conceived and designed the study. S.G.D., H.H., and S.A.L. obtained funding to support the clinical enrollment of study participants, and S.A.L. and SGD obtained funding to support the characterization of the HIV reservoir. S.E.C., S.B., D.H., and M.G. facilitated coordination with the San Francisco Department of Public Health and Ward 86 clinical services to link patients into care and provided critical feedback on the clinical management of acute HIV. S.A.L., R.H., S.G.D., T.J.H., H.M.H., S.S., S.D., and V.P. coordinated the collection, management, and quality control processes for the clinical data, and S.A.L., S.G.D., J.M., F.H., and C.P. provided biospecimens. J.M. and T.J.H. performed biospecimen processing, and G.M.L., M.M., and K.R. performed the HIV reservoir assays. L.S. developed the initial decay models under the guidance of S.A.L. and J.W., and A.B. and S.A.L. further modified these models under the guidance of J.W., L.S., and J.S. G.M.L., M.M., K.R., L.S., J.S., J.W., and S.A.L. analyzed the HIV reservoir data. A.B., S.A.L., L.S., J.S., J.W., S.S., S.D., V.P., and C.S. performed data visualization for the manuscript. S.A.L. and A.B. wrote the report with critical feedback from L.S. J.W. and the additional authors. Correspondence should be addressed to S.A.L. or A.B.

## Competing interests

The authors declare no competing interests.
