## [Transparent Peer Review file · Nature Communications]

Rapid Biphasic Decay of Intact and Defective HIV DNA Reservoir During Acute Treated HIV Disease

Corresponding Author: Dr Sulggi Lee

Version 0:

Reviewer comments:

Reviewer #1

(Remarks to the Author)

This is a nice study reporting estimates of decay rates of HIV DNA in the acute phase of HIV infection. The study uses a novel dataset, of longitudinal measurements from 67 individuals initiated with ART within 100 days of infection. Using IPDA on peripheral CD4 T cells, the levels of intact and defective proviruses as a function of time from treatment initiation was measured. The data was analyzed using a nonlinear generalized additive modelling framework to estimate decay rates which controlling for factors such as pre-treatment viral load, time of treatment onset, baseline CD4 counts, etc. The study found that a biphasic decay model best described the data, with a fast first phase decline followed by a slower second phase decline. Interestingly, there seemed not to be much of a difference between intact and defective proviral compartments. Further, the timing of treatment initiation affected the decay rates, with early initiation leading to faster decay. Further, higher baseline CD4 counts and lower baseline viral loads also led to increased decay rates. These findings are interesting and important and have implications of HIV cure studies. The dataset is unique. The analysis is nicely done. The paper is well written. I have a few comments for the authors to consider.

Comments:

1. The authors find that the first phase decay rates they estimate are similar to previous estimates of the decay rates when the treatment was initiated in the chronic phase. However, the second phase decay rates in the present study are five times higher than the chronic phase scenarios. The authors mention in the discussion that the second phase is attributed to the conversion of activated cells to memory cells, the contraction phase. Is there thus evidence that faster conversion to memory happens in the acute phase than in the chronic phase? Is there evidence from primate models where this is tested? If not, what other hypotheses could explain the differences?
2. The authors mention that the first phase decay rates of the intact and defective compartments are comparable. However, they seem to be a factor of 2 different: half-life of 0.7 months for intact vs 0.34 months for defective, as listed in Page 10. Thus, I am not sure of the assertion of similarity. The authors must make the assertion with a formal statistical test. Further, as the authors mention later, one expects the intact viral compartment to decay faster than the defective ones because of a greater chance of immune recognition of the former. The numbers above seem to suggest the opposite. The authors should clarify.
3. The inflection point of 5 weeks is intriguing. The authors do not discuss its potential origins. We do recognize that during the course of ART many transitions from one phase to the next happen. Yet, this first transition, especially given that the half-life of the first phase decay is the same in acute and chronic settings, would be interesting to understand. When does the inflection happen in the chronic setting? It would help for the authors to compare the two and at least speculate on the origins in the acute setting.

Reviewer #2

(Remarks to the Author)

Barbehenn et al., developed and applied a mathematical model to predict the decay of defective and intact provirus in a cohort of 67 people with HIV (PWH) treated during acute infection using intact proviral DNA assay (IPDA). As previously reported for PWH treated during chronic infection, they found a biphasic decay after ART initiation of both Intact and defective proviral DNA, with rapid initial decline followed by a slower decay. The decay of proviral DNA was 5-fold faster in PWH treated during acute infection as compared to the estimates in PWH treated during chronic infection. Interestingly, they found that faster proviral DNA decay was associated with clinical parameters such as high initial CD4 count, and lower pre-

ART viral load. The topic is important, the analyses is carefully performed and the conclusions are supported by the data. However, the study seems rather preliminary in a way that it does not advance enough our current knowledge in the field. Indeed, the reservoir dynamics has been largely described in PWH treated during chronic phase of infection but not just. HIV reservoir dynamics has also been addressed in PWH treated during acute infection. Particularly, Massanella et al addressed this topic by measuring total, integrated and 2LTR circle viral DNA. While in the present manuscript IPDA was used to distinguish between defective and intact proviruses, TILDA, which measures inducible/functional HIV reservoir cells was used in Massanella et al. This reviewer was surprised that the authors did not discuss this manuscript. Additionally, a recent manuscript by De Clercq et al. has also addressed HIV reservoir dynamics in PWH treated during acute infection among other immunological questions (De Clercq et al. *Frontiers in immunology* 2024).

By measuring proviral DNA (intact and defective) decay starting from the time of ART initiation, the authors are measuring the decay of productively infected cells rather than long lasting latent reservoir cells in blood CD4 T cells. Particularly, at time of ART initiation during the acute phase of infection, productively infected CD4 T cells are by far more represented than the latent HIV reservoir cells that persist on long term ART.

In conclusion, with the exception of the finding of the link between proviral DNA decay and initial CD4 count, and pre-ART viral load, the manuscript lack important novelty to justify its publication in Nature communication.

Version 1:

Reviewer comments:

Reviewer #1

(Remarks to the Author)

The authors have done a nice job of addressing my comments. I have no further comments.

Reviewer #2

(Remarks to the Author)

The authors responded to my concerns by including and discussing important manuscripts published earlier by others. In this regard, I'm satisfied. However, I still questioning the novelty of their findings and its potential impact on our understanding of the establishment and dynamics of the long lasting latent reservoir cells in blood CD4 T.

Reviewer #1 (Remarks to the Author):

1. The authors find that the first phase decay rates they estimate are similar to previous estimates of the decay rates when the treatment was initiated in the chronic phase. However, the second phase decay rates in the present study are five times higher than the chronic phase scenarios. The authors mention in the discussion that the second phase is attributed to the conversion of activated cells to memory cells, the contraction phase. Is there thus evidence that faster conversion to memory happens in the acute phase than in the chronic phase? Is there evidence from primate models where this is tested? If not, what other hypotheses could explain the differences?

We appreciate the reviewer's request for additional clarification on our observed first (week 0-5) and second (week 5-24) phases of decay. Indeed, we observed similar first phase of decay but faster second phase of decay of HIV intact DNA compared to one prior study of chronic treated PWH. Several prior studies have now clearly demonstrated that this first phase of decay is likely due to death of productively infected cells;¹⁻⁵ we believe that a similar mechanism is likely driving the initial rapid decay observed in both acute and chronic PWH initiating ART.

Our work suggests that effectiveness of the host-mediated clearance of the reservoir during the second phase differs in those treated early versus late. There has only been one study modeling HIV intact DNA decay rates after ART initiation among chronic PWH (White et. al.)¹ and other than our study none modeling HIV intact DNA decay rates after ART initiation among acute PWH; there are no primate models directly comparing acute versus chronic treated SIV intact DNA

decay rates. As this is a new finding, there are no direct experimental data which might account for our observed differences. Based on prior experimental work, the second phase is thought to represent a “contraction phase” when activated cells transition from an effector to a memory phenotype as antigen decreases with ART initiation.⁶⁻⁸ Thus, the decay rate during this contraction phase is probably largely driven by decay of these longer-lived memory cells. A recent study by Fray et. al.,⁹ (using a modified IPDA) described SIV intact DNA decay rates similar to that reported by White et. al. among chronic treated PWH; these data, while not directly comparing to decay rates in acute treated SIV, further support the limited data describing slower second phase decay rates in chronic treated HIV/SIV compared to in our acute treated HIV population. We hypothesize that the reasons for the faster second phase of HIV intact DNA decay in our study may be due to real biological differences (e.g., perhaps faster conversion from activated to memory state, as the reviewer queries; acute PWH have been shown to have less exhausted and more functional immune responses than chronic PWH,¹⁰⁻¹² perhaps allowing a more effective innate immune responses) or due to differences in study design (e.g., perhaps our frequent early sampling – every 2-4 weeks – allowed greater precision in our estimated $t_{1/2}$ of this second phase of decay compared to the study by White et. al., which had much less frequent sampling). Further work are clearly needed to add to the limited literature on HIV reservoir decay rates during acute versus chronic infection.

We have now added further clarification of this in the Discussion (lines 243-269).

2. The authors mention that the first phase decay rates of the intact and defective compartments are comparable. However, they seem to be a factor of 2 different: half-life of 0.7 months for intact vs 0.34 months for defective, as listed in Page 10. Thus, I am not sure of the assertion of similarity. The authors must make the assertion with a formal statistical test. Further, as the authors mention later, one expects the intact viral compartment to decay faster than the defective ones because of a greater chance of immune recognition of the former. The numbers above seem to suggest the opposite. The authors should clarify.

We apologize for the oversight carried over from a former version of the draft (our final model *did indeed demonstrate statistical significance*, suggesting that HIV *defective* DNA decayed faster than HIV intact DNA during the first phase). Our observation is in contrast to prior studies of PWH on long-term ART where HIV *intact* DNA has been shown to decay faster than defective DNA (thought to largely be driven by selective immune pressure during ART).¹³⁻¹⁶ However, even these studies have been inconsistent, with some individuals demonstrating a plateau or even an increase in HIV intact DNA after prolonged (e.g., 20 years) ART.^{14,15} Thus, while our data suggests a faster decay of HIV defective DNA compared to intact DNA, it remains unclear if this observation reflects **(1) true biological phenomena** uniquely captured by our frequent early sampling and/or **(2) unique properties of the IPDA**.

It has been well described that there are at least three groups of HIV-infected cells in PWH: **(i) truly “intact” proviruses**, **(ii) “partially defective” proviruses** that can produce defective HIV RNA/proteins, which, despite being unable to produce virus, can still lead to immunogenic/cytopathic effects,¹⁷ and **(iii) truly “inert” proviruses** that express no HIV RNA or proteins. It is possible that by quantifying defective DNA decay rates at very early, frequently sampled timepoints

using the IPDA, we were able to capture true biologic phenomena – i.e., the decay of both truly intact (i) and partially defective (ii) proviruses – recognized and cleared by the host’s early immune response to HIV. This would be consistent with findings from two prior acute HIV cohorts which did not measure IPDA (intact and defective DNA) but did quantify HIV total DNA, integrated DNA, and 2-LTR DNA.^{18,19} While these other assays do not discriminate between intact vs. defective proviruses, they do estimate the most common form of HIV-infected cells during acute infection: highly unstable unintegrated linear HIV DNA (with an estimated half-life of ~2 days).²⁰ Since the majority of the HIV reservoir consists of defective provirus,²¹ estimates of HIV total, integrated, and 2-LTR DNA decay rates from these other studies ($t_{1/2}$ = 14.5, 14.1, and 30.5 days, respectively¹⁸) are largely consistent with our estimates of defective DNA decay rates ($t_{1/2}$ =9.5 days) during the first phase of decay, suggesting a potential true biological phenomena that warrants further study. Alternatively, a second possibility is that our observations reflect some misclassification of HIV “intact” provirus (i.e., since the IPDA targets just two regions of the HIV genome to define “defective” provirus²²). However, Reeves and colleagues recently performed detailed validation experiments (e.g., using quantitative viral outgrowth assay and near full-length sequencing) and showed that the rate of misclassification is <5% with the IPDA,²³ suggesting that this degree of misclassification alone would be unlikely to fully explain our findings.

We have now revised our Results (lines 174-178) and Discussion (lines 296-309) to clarify our findings and the limitations around these observations. More importantly, these data highlight the need to apply these modeling approaches (and frequent early sampling timepoints) in other acute HIV cohorts to confirm our novel findings.

3. The inflection point of 5 weeks is intriguing. The authors do not discuss its potential origins. We do recognize that during the course of ART many transitions from one phase to the next happen. Yet, this first transition, especially given that the half-life of the first phase decay is the same in acute and chronic settings, would be interesting to understand. When does the inflection happen in the chronic setting? It would help for the authors to compare the two and at least speculate on the origins in the acute setting.

The determination of the inflection point is a key finding of our study. Indeed, the reason for our mathematical modeling is to leverage the large number of longitudinal samples to allow the data to identify a potential biologically meaningful inflection point. We have now improved upon our previously (more heavily statistical) description to help a broader audience follow our comprehensive modeling approach: (1) we fit various models – monophasic, biphasic, triphasic (Figure 2), (2) we iteratively performed fine tuning of our data to choose the best fit value for tau (τ) – i.e., the inflection point(s) (Figure 3), and (3) we included key clinical predictors associated with HIV reservoir size and/or decay (initial CD4+ T cell count, pre-ART viral load, timing of ART initiation). We also fit both linear and nonlinear models to flexibly allow our data to inform the selection of the best fit model (mono-, bi-, or triphasic) and define the best fit inflection point(s). In White et al., chronic-treated PWH had an estimated intact HIV DNA reservoir inflection point of ~8-11 weeks; this is a longer first phase of decay than we estimated in our acute-treated population (~5 weeks). The longer first phase of decay for chronic-treated PWH may be driven by their higher initial intact HIV DNA reservoir size or may be a side effect

of the modeling artifact accounting for the much slower decay during the second phase. The overall goal of this work is to develop a comprehensive mathematical model that can be readily adapted to global acute HIV cohorts so that these findings can be validated and/or identify key differences by population. We have now added further clarification of this in our Results (lines 151-168) and Discussion (lines 229-242; 273-287).

As an additional measure in response to the reviewer, we feel that it will greatly add rigor to the quality of our paper to demonstrate how this modeling approach can be applied to other measures. In this revised submission, we have taken the time to add in new data, applying the same mathematical modeling approach to plasma HIV RNA (measured at each study visit using the Abbott Real Time PCR assay, limit of detection < 40 copies/mL). We again fit various mono-, bi-, and triphasic models for plasma HIV RNA, and our model fit a triphasic decay pattern with predicted inflection points at 0.5 and 4 weeks (Supplementary Fig. 11). Our model was not only consistent with prior published work describing triphasic decay of plasma HIV RNA in treatment naïve PWH initiating integrase inhibitor-based therapy,^{2,24} given that plasma (cell-free) HIV RNA correlates with productively infected cells, we found that our plasma HIV RNA decay estimates were consistent with previously reported clearance rates of productively infected CD4+ T cells in the literature ($t_{1/2} \sim 0.7$ days).²⁰ These new data have been added to our Results (lines 208-226) and Discussion (lines 235-236; 257-261).

Reviewer #2 (Remarks to the Author):

4. Barbehenn et al., developed and applied a mathematical model to predict the decay of defective and intact provirus in a cohort of 67 people with HIV (PWH) treated during acute infection using intact proviral DNA assay (IPDA)... Interestingly, they found that faster proviral DNA decay was associated with clinical parameters such as high initial CD4 count, and lower pre-ART viral load. The topic is important, the analyses is carefully performed and the conclusions are supported by the data. However, the study seems rather preliminary in a way that it does not advance enough our current knowledge in the field. Particularly, Massanella et al addressed this topic by measuring total, integrated and 2LTR circle viral DNA. While in the present manuscript IPDA was used to distinguish between defective and intact proviruses, TILDA, which measures inducible/functional HIV reservoir cells was used in Massanella et al. This reviewer was surprised that the authors did not discuss this manuscript.

While we greatly appreciate the reviewer's suggestion to include this important paper, we respectfully disagree as to the novelty of our work. We have now included the paper by Massanella et. al., as well as the one by De Clercq et. al., in our citations – but we have also included critical discussion around key differences in these prior studies and ours in the Discussion (lines 270-287).

There are limited studies to date describing HIV intact DNA decay rates after ART initiation among acute PWH. The study by Massanella et. al.,¹⁸ included frequent sampling after ART initiation among acute treated PWH in Peru, but did not measure HIV intact and defective DNA, precluding direct comparison to our results. However, importantly, they reported similar biphasic patterns as described in our study, using other PCR-based assays (HIV total, integrated, and 2-LTR DNA). Leyre et. al.,¹⁹ quantified these same PCR-based assays (HIV total, integrated, and 2-LTR DNA) in another cohort of acute

treated PWH in Thailand, and while they did not perform formal mathematical modeling, clearly show in their figures that these measures declined quickly at first, followed by a slower decline. Other acute HIV studies measuring HIV reservoir changes after ART initiation – whether sampling less frequently,^{25,26} or using different assays than the IPDA,^{18,19,25} or not including formal mathematical modeling^{19,25,26} – are now mentioned in our paper. However, key differences in HIV reservoir assays and sampling frequency make direct comparisons difficult with ours.

Similarly, there is only one study to date describing HIV intact DNA decay rates after ART initiation among chronic PWH (White et. al.; similar decay rates in chronic treated SIV recently described by Fray et. al.). This is likely because enrolling and sampling chronic PWH at the time of ART initiation (especially at the frequent sampling timepoints as performed here) has been historically, less common. Indeed, in our well characterized SCOPE HIV cohort, the vast majority of our chronic HIV participants were enrolled into study long after HIV diagnosis and after ART initiation.

None of these prior studies, including the two that included mathematical modeling of the reservoir decay,^{1,18} controlled for key clinical predictors that may further modify decay rates (e.g., initial CD4+ T cell counts, pre-ART viral load, timing of ART initiation).

However, while the study by Massanella and colleagues did not measure IPDA (HIV intact and defective DNA) decay rates, and are thus not directly comparable to our findings, their study did model the decay rates of HIV total, integrated DNA, and 2-LTR DNA, as well as the quantitative viral outgrowth assay (QVOA²⁷⁻²⁹) and the multiply spliced tat/rev assay (TILDA³⁰). The QVOA and TILDA were less scalable assays for the large number of samples analyzed here; these assays require a large number of input cells, and quantification of these assays is higher than for PCR-based assays and follow Poisson statistics,³¹ leading to underestimation of the size of the replication-competent reservoir.²¹ Perhaps for these reasons, the authors were unable to fit decay models for QVOA or TILDA (due to the low frequency of infected cells). However, it is notable that the authors did observe biphasic decay patterns for HIV total, integrated, and 2-LTR DNA with very similar inflection points (6 weeks) as described in our models (5 weeks).

Finally, it is important to note that the decay rates described here are complementary to, but not the same as, decay rates described in several long-term ART studies.^{13,14,16,28,32,33} In these long-term ART studies, the objective was to describe decay rates long after ART initiation, often out to 20 years of viral suppression. Overall, the studies found that HIV intact DNA decayed more quickly than defective DNA, with another inflection point at ~7 years of ART.¹³⁻¹⁶ For these reasons, our data provide for the first time additional granularity of HIV intact DNA decay rates during the earliest stages of acute treated HIV. While during long-term ART suppression, HIV intact DNA likely decays faster than defective DNA due to selective immune clearance of cells harboring intact (“replication-competent”) virus which are more readily detected by host immune surveillance,^{13-16,34,35} it remains unclear whether selective immune clearance plays a major role during acute treated HIV (see above response #1 to Reviewer 1). The initial phase of HIV intact decay described in our study suggests further work is needed to clarify the underlying reasons for potential differential decay of HIV intact and defective DNA.

For these reasons, there is a critical need to confirm our findings in other cohorts with similar study designs and scalable assays that will allow broadly generalizable to a global population of PWH. This includes accounting for differences in

HIV reservoir decay rates that may be influenced by host genetics (i.e., ancestry), viral strain (HIV subtype), participant demographics (e.g., sex, race/ethnicity), or clinical guidelines (e.g., options for initial ART regimens, clinical indications for initiating ART). Defining complex reservoir decay dynamics at the time of HIV reservoir establishment/stabilization (i.e., ART initiation), including how key host factors such as the timing of ART initiation, initial CD4+ T cell count, and pre-ART viral load influence these decay rates, will be necessary to inform future therapies that can equitably benefit all PWH. For example, chronic treated PWH may benefit from the same curative interventions as acute treated PWH but might require adjunctive therapies to reverse cell exhaustion.³⁶⁻³⁸

5. Additionally, a recent manuscript by De Clercq et al. has also addressed HIV reservoir dynamics in PWH treated during acute infection among other immunological questions (De Clercq et al. *Frontiers in immunology* 2024).

We appreciate the reviewer's suggestion and have now included the paper by De Clercq and colleagues in our citations, as well as in our Discussion. In their study, De Clercq et. al., performed IPDA measurements from N=37 acute treated PWH. While their study included acute treated PWH, the authors did not include any statistical modeling of HIV intact or defective DNA decay rates; data are only presented as Wilcoxon comparisons or Spearman correlations. Furthermore, sampling to support their conclusion that “the intact HIV reservoir declines faster than the total HIV reservoir” included four broad timepoints “T0, DVL, UD, and UD+1” where T0 = during acute HIV infection; DVL = decreasing viral load on ART; UD = after viral suppression of plasma viral load (median 278 days); and UD+1 = ~1 year later (median 726 days).

Instead, the primary focus of the study by De Clercq et. al., appeared to be the analysis of 40 plasma cytokines, for which 8 proinflammatory cytokines previously noted to be elevated in acute PWH were found to be significantly higher in their acute treated HIV group compared to the HIV-uninfected group. Notably, these cytokines were determined to be statistically significant without correction for multiple testing. Thus, these cytokine analyses are also in contrast to recently presented data from our acute HIV cohort (Barbehenn et. al., Conference on Retroviruses and Opportunistic Infections, Abstract #508). Using our biphasic decay models of HIV intact and defective DNA, from our cohort, we observed two statistically significant plasma cytokines associated with accelerated decay of HIV intact DNA (during the first phase of decay): interleukin-10 and interferon-beta (a type I IFN).

For these reasons, while we have now included mention of the study by De Clercq et. al. in our manuscript, we strongly feel that our detailed mathematical modeling is novel and adds to the limited literature describing HIV reservoir decay rates – especially the intact HIV DNA reservoir – during acute treated HIV. Dissemination of our findings, and the potential utility of these mathematical models, are critical for encouraging future collaborative efforts across global acute HIV cohorts. This type of systematic approach may be an important step in disentangling cohort-specific effects (e.g, due to host genetic ancestry, HIV-1 subtype strain, clinical practices in initiating immediate ART, etc.) potentially informing future personalized HIV cure strategies to benefit all PWH.

6. By measuring proviral DNA (intact and defective) decay starting from the time of ART initiation, the authors are measuring the decay of productively infected cells rather than long lasting latent reservoir cells in blood CD4 T cells. Particularly, at time of ART initiation during the acute phase of infection, productively infected CD4 T cells are by far more represented than the latent HIV reservoir cells that persist on long term ART. In conclusion, with the exception of the finding of the link between proviral DNA decay and initial CD4 count, and pre-ART viral load, the manuscript lack important novelty to justify its publication in Nature communication.

We respectfully disagree with the reviewer and reference our detailed responses above for Q#2 and Q#4. The HIV reservoir is largely established/stabilized at the time of ART initiation.³⁹ While we agree that the long-lived latent reservoir is a key defining target for HIV cure, how and where these cells then become “the long lasting latent reservoir” remains unclear. Even in long-term studies of reservoir decay out to 20 years suggests that the decay patterns are not broadly generalizable (with some individuals even showing an increase or plateau of HIV intact DNA towards these longer durations of therapy).^{14,15} Here, we advocate for a scalable approach that allows cross-cohort collaboration – ideally, aligning study designs and assays across cohorts, and performing comprehensive modeling of decay rates of the HIV reservoir. As discussed in our response to Q#2 above, there are at least three populations of infected cells in PWH: (1) those that express viral RNA and proteins, representing the “intact” reservoir (undergo selective clearance and are thus, the main driver of reservoir decay during long-term suppressive ART); (2) those that express no viral RNA or proteins, truly “inert” provirus-containing cells (whether from acute or chronic PWH); and (3) those that make defective partial viral RNA and proteins (do not produce virus but are still immunogenic/cytopathic¹⁷). We advocate for scalable assays that can broaden our understanding of HIV reservoir decay patterns that may be applicable and accessible to a global population of people living with HIV; this does not preclude the ability to study more refined HIV reservoir assays (but such assays may be limited by sample availability and/or cost, and may detect low frequencies of HIV-infected cells, limiting the ability to precisely model decay rates).

We appreciate this opportunity to publish this important work in *Nature Communications*. Thank you again for considering this manuscript for publication.

Sincerely,

Alton Barbehenn, Ph.D.

Sulggi A. Lee, M.D., Ph.D.

Literature Cited

- 1 White, J. A. *et al.* Complex decay dynamics of HIV virions, intact and defective proviruses, and 2LTR circles following initiation of antiretroviral therapy. *Proc Natl Acad Sci U S A* **119** (2022). <https://doi.org:10.1073/pnas.2120326119>
- 2 Andrade, A. *et al.* Three distinct phases of HIV-1 RNA decay in treatment-naive patients receiving raltegravir-based antiretroviral therapy: ACTG A5248. *J Infect Dis* **208**, 884-891 (2013). <https://doi.org:10.1093/infdis/jit272>
- 3 Murray, J. M., Kelleher, A. D. & Cooper, D. A. Timing of the components of the HIV life cycle in productively infected CD4+ T cells in a population of HIV-infected individuals. *J Virol* **85**, 10798-10805 (2011). <https://doi.org:10.1128/JVI.05095-11>
- 4 Gilmore, J. B., Kelleher, A. D., Cooper, D. A. & Murray, J. M. Explaining the determinants of first phase HIV decay dynamics through the effects of stage-dependent drug action. *PLoS Comput Biol* **9**, e1002971 (2013). <https://doi.org:10.1371/journal.pcbi.1002971>
- 5 Perelson, A. S. *et al.* Decay characteristics of HIV-1-infected compartments during combination therapy. *Nature* **387**, 188-191 (1997). <https://doi.org:10.1038/387188a0>
- 6 Shan, L. *et al.* Transcriptional Reprogramming during Effector-to-Memory Transition Renders CD4(+) T Cells Permissive for Latent HIV-1 Infection. *Immunity* **47**, 766-775 e763 (2017). <https://doi.org:10.1016/j.immuni.2017.09.014>
- 7 De Boer, R. J., Homann, D. & Perelson, A. S. Different dynamics of CD4+ and CD8+ T cell responses during and after acute lymphocytic choriomeningitis virus infection. *J Immunol* **171**, 3928-3935 (2003). <https://doi.org:10.4049/jimmunol.171.8.3928>
- 8 Zhan, Y., Carrington, E. M., Zhang, Y., Heinzl, S. & Lew, A. M. Life and Death of Activated T Cells: How Are They Different from Naive T Cells? *Front Immunol* **8**, 1809 (2017). <https://doi.org:10.3389/fimmu.2017.01809>
- 9 Fray, E. J. *et al.* Antiretroviral therapy reveals triphasic decay of intact SIV genomes and persistence of ancestral variants. *Cell Host Microbe* **31**, 356-372 e355 (2023). <https://doi.org:10.1016/j.chom.2023.01.016>
- 10 Takata, H. *et al.* Long-term antiretroviral therapy initiated in acute HIV infection prevents residual dysfunction of HIV-specific CD8(+) T cells. *EBioMedicine* **84**, 104253 (2022). <https://doi.org:10.1016/j.ebiom.2022.104253>
- 11 Oxenius, A. *et al.* Early highly active antiretroviral therapy for acute HIV-1 infection preserves immune function of CD8+ and CD4+ T lymphocytes. *Proc Natl Acad Sci U S A* **97**, 3382-3387 (2000). <https://doi.org:10.1073/pnas.97.7.3382>
- 12 Streeck, H. *et al.* Immunological and virological impact of highly active antiretroviral therapy initiated during acute HIV-1 infection. *J Infect Dis* **194**, 734-739 (2006). <https://doi.org:10.1086/503811>
- 13 Peluso, M. J. *et al.* Differential decay of intact and defective proviral DNA in HIV-1-infected individuals on suppressive antiretroviral therapy. *JCI Insight* **5** (2020). <https://doi.org:10.1172/jci.insight.132997>
- 14 Gandhi, R. T. *et al.* Varied Patterns of Decay of Intact Human Immunodeficiency Virus Type 1 Proviruses Over 2 Decades of Antiretroviral Therapy. *J Infect Dis* **227**, 1376-1380 (2023). <https://doi.org:10.1093/infdis/jiad039>
- 15 McMyn, N. F. *et al.* The latent reservoir of inducible, infectious HIV-1 does not decrease despite decades of antiretroviral therapy. *J Clin Invest* **133** (2023). <https://doi.org:10.1172/JCI171554>
- 16 Gandhi, R. T. *et al.* Selective Decay of Intact HIV-1 Proviral DNA on Antiretroviral Therapy. *J Infect Dis* **223**, 225-233 (2021). <https://doi.org:10.1093/infdis/jiaa532>
- 17 Pollack, R. A. *et al.* Defective HIV-1 Proviruses Are Expressed and Can Be Recognized by Cytotoxic T Lymphocytes, which Shape the Proviral Landscape. *Cell Host Microbe* **21**, 494-506 e494 (2017). <https://doi.org:10.1016/j.chom.2017.03.008>
- 18 Massanella, M. *et al.* Long-term effects of early antiretroviral initiation on HIV reservoir markers: a longitudinal analysis of the MERLIN clinical study. *Lancet Microbe* **2**, e198-e209 (2021). [https://doi.org:10.1016/s2666-5247\(21\)00010-0](https://doi.org:10.1016/s2666-5247(21)00010-0)
- 19 Leyre, L. *et al.* Abundant HIV-infected cells in blood and tissues are rapidly cleared upon ART

- initiation during acute HIV infection. *Sci Transl Med* **12** (2020).
<https://doi.org/10.1126/scitranslmed.aav3491>
- 20 Simon, V. & Ho, D. D. HIV-1 dynamics in vivo: implications for therapy. *Nat Rev Microbiol* **1**, 181-190 (2003). <https://doi.org/10.1038/nrmicro772>
 - 21 Ho, Y. C. *et al.* Replication-competent noninduced proviruses in the latent reservoir increase barrier to HIV-1 cure. *Cell* **155**, 540-551 (2013). <https://doi.org/10.1016/j.cell.2013.09.020>
 - 22 Kinloch, N. N. *et al.* HIV-1 diversity considerations in the application of the Intact Proviral DNA Assay (IPDA). *Nat Commun* **12**, 165 (2021). <https://doi.org/10.1038/s41467-020-20442-3>
 - 23 Reeves, D. B. *et al.* Impact of misclassified defective proviruses on HIV reservoir measurements. *Nat Commun* **14**, 4186 (2023). <https://doi.org/10.1038/s41467-023-39837-z>
 - 24 Palmer, S. *et al.* Low-level viremia persists for at least 7 years in patients on suppressive antiretroviral therapy. *Proc Natl Acad Sci U S A* **105**, 3879-3884 (2008). <https://doi.org/10.1073/pnas.0800050105>
 - 25 De Clercq, J. *et al.* Longitudinal patterns of inflammatory mediators after acute HIV infection correlate to intact and total reservoir. *Front Immunol* **14**, 1337316 (2023).
<https://doi.org/10.3389/fimmu.2023.1337316>
 - 26 Reddy, K. *et al.* Differences in HIV-1 reservoir size, landscape characteristics and decay dynamics in acute and chronic treated HIV-1 Clade C infection. *eLife* **13** (2024).
<https://doi.org/10.1101/https://doi.org/10.7554/eLife.96617.11/2024.02.16.24302713>
 - 27 Chun, T. W. *et al.* Early establishment of a pool of latently infected, resting CD4(+) T cells during primary HIV-1 infection. *Proc Natl Acad Sci U S A* **95**, 8869-8873 (1998).
<https://doi.org/10.1073/pnas.95.15.8869>
 - 28 Finzi, D. *et al.* Identification of a reservoir for HIV-1 in patients on highly active antiretroviral therapy. *Science* **278**, 1295-1300 (1997). <https://doi.org/10.1126/science.278.5341.1295>
 - 29 Siliciano, J. D. *et al.* Long-term follow-up studies confirm the stability of the latent reservoir for HIV-1 in resting CD4+ T cells. *Nat Med* **9**, 727-728 (2003). <https://doi.org/10.1038/nm880>
 - 30 Procopio, F. A. *et al.* A Novel Assay to Measure the Magnitude of the Inducible Viral Reservoir in HIV-infected Individuals. *EBioMedicine* **2**, 872-881 (2015).
<https://doi.org/10.1016/j.ebiom.2015.06.019>
 - 31 Eriksson, S. *et al.* Comparative analysis of measures of viral reservoirs in HIV-1 eradication studies. *PLoS pathogens* **9**, e1003174 (2013). <https://doi.org/10.1371/journal.ppat.1003174>
 - 32 Antar, A. A. *et al.* Longitudinal study reveals HIV-1-infected CD4+ T cell dynamics during long-term antiretroviral therapy. *J Clin Invest* **130**, 3543-3559 (2020). <https://doi.org/10.1172/JCI135953>
 - 33 Siliciano, J. D. *et al.* Long-term follow-up studies confirm the stability of the latent reservoir for HIV-1 in resting CD4+ T cells. *Nature medicine* **9**, 727-728 (2003). <https://doi.org/10.1038/nm880>
 - 34 Pinzone, M. R. *et al.* Longitudinal HIV sequencing reveals reservoir expression leading to decay which is obscured by clonal expansion. *Nat Commun* **10**, 728 (2019). <https://doi.org/10.1038/s41467-019-08431-7>
 - 35 Gondim, M. V. P. *et al.* Heightened resistance to host type 1 interferons characterizes HIV-1 at transmission and after antiretroviral therapy interruption. *Sci Transl Med* **13** (2021).
<https://doi.org/10.1126/scitranslmed.abd8179>
 - 36 Gay, C. L. *et al.* Clinical Trial of the Anti-PD-L1 Antibody BMS-936559 in HIV-1 Infected Participants on Suppressive Antiretroviral Therapy. *J Infect Dis* **215**, 1725-1733 (2017).
<https://doi.org/10.1093/infdis/jix191>
 - 37 Gay, C. L. *et al.* Suspected Immune-Related Adverse Events With an Anti-PD-1 Inhibitor in Otherwise Healthy People With HIV. *J Acquir Immune Defic Syndr* **87**, e234-e236 (2021).
<https://doi.org/10.1097/QAI.0000000000002716>
 - 38 Colston, E. *et al.* An open-label, multiple ascending dose study of the anti-CTLA-4 antibody ipilimumab in viremic HIV patients. *PLoS One* **13**, e0198158 (2018).
<https://doi.org/10.1371/journal.pone.0198158>
 - 39 Abrahams, M. R. *et al.* The replication-competent HIV-1 latent reservoir is primarily established near the time of therapy initiation. *Sci Transl Med* **11** (2019).
<https://doi.org/10.1126/scitranslmed.aaw5589>

Sulggi A. Lee, MD, PhD
Associate Professor of Medicine

UCSF Pride Hall, 2540 23rd Street,
Room 4211, Box 1272
San Francisco, CA 94110
Tel.: (415) 735-5127
Fax.: (415) 476-6953
Email: sulggi.lee@ucsf.edu

October 4, 2024

Thank you for the subsequent review of our manuscript “Rapid Biphasic Decay of Intact and Defective HIV DNA Reservoir During Acute Treated HIV Disease.” for publication in *Nature Communications*. We greatly appreciate the opportunity to further improve the manuscript. We have responded to the additional reviewer comments below.

Reviewer #1 (Remarks to the Author):

The authors have done a nice job of addressing my comments. I have no further comments.

We appreciate the reviewer’s comment.

Reviewer #2 (Remarks to the Author):

The authors responded to my concerns by including and discussing important manuscripts published earlier by others. In this regard, I’m satisfied. However, I still questioning the novelty of their findings and its potential impact on our understanding of the establishment and dynamics of the long lasting latent reservoir cells in blood CD4 T.

We appreciate the reviewer’s comment and have removed any statements that include the terms “first”, “new”, “novel”, etc.

Thank you so much again for your consideration and careful review. We believe that the manuscript is now greatly improved because of the reviewers’ helpful and comprehensive suggestions. We hope that these revisions are acceptable and allow publication in *Nature Communications*.

Sincerely,

Alton Barbehenn, Ph.D.

Sulggi A. Lee, M.D., Ph.D.